# Pharmacological vs. Non-Pharmacological Treatment in the Management of Relative Energy Deficiency in Sport (REDs): A Systematic Review and Meta-Analysis

**DOI:** 10.3390/sports13120453

**Published:** 2025-12-15

**Authors:** Aimee Nicole Wood, Andrew Soundy

**Affiliations:** Department of Physiotherapy, School of Sport, Exercise and Rehabilitation Sciences, University of Birmingham, Birmingham B15 2TS, UK; a.a.soundy@bham.ac.uk

**Keywords:** Relative Energy Deficiency in Sport (REDs), intervention, pharmacological, non-pharmacological, energy availability

## Abstract

Objective: The purpose of the study was to conduct a systematic review assessing the impact of pharmacological and non-pharmacological interventions on Relative Energy Deficiency in Sport (REDs). The study design follows a systematic review and meta-analysis. The data sources are CINAHL, MEDLINE, SportDiscus, ERIC, and Embase from inception until July 2025. Eligibility criteria for selecting the studies include the experimental, quasi-experimental, and pre-experimental literature that investigated interventions designed to support the symptoms of REDs. Results: A total of nineteen studies (fifteen non-pharmacological interventions, four pharmacological interventions), with a total of 759 females, were included in the review. Non-pharmacological interventions demonstrated positive benefits on menstrual function recovery, energy availability, fat mass, and body fat percentage. Meta-analyses quantified nutrition intervention benefits on an individual’s fat mass (kg), 1.36 (95% CI 0.68, 2.04), and body percentage fat (%), 2.21 (95% CI 1.34, 3.08). It was also possible to identify the impact of non-pharmacological interventions on total triiodothyronine (T3) biomarkers (nmol/L), −2.37 (95% CI −5.57, 0.83). It should be noted, however, that non-pharmacological interventions were limited by quality and certainty assessment, identifying included evidence as low to moderate. Pharmacological interventions demonstrated some positive (at times very strong effect sizes) results for impact on bone mineral density, but conclusions are currently limited by well-powered experimental studies. Conclusions: The current evidence base favors non-pharmacological management as an initial response for managing REDs. Initial pharmacological management appears to identify limited but potentially (depending on the drug) promising evidence for the impact on bone mineral density; further evidence is required to be more certain about the impact on hormonal profiling and menstrual recovery function. Further research is needed to help develop a greater understanding.

## 1. Introduction

Relative Energy Deficiency in Sport (REDs) is a relatively new and evolving clinical model that is defined as “*A syndrome of impaired physiological and/or psychological functioning experienced by female and male athletes that is caused by exposure to problematic (prolonged and/or severe) low energy availability (LEA). The detrimental outcomes include, but are not limited to, decreases in energy metabolism, reproductive function, musculoskeletal health, immunity, glycogen synthesis and cardiovascular and haematological health, which can all individually and synergistically lead to impaired well-being, increased injury risk and decreased sports performance*” [1]. LEA results when the energy intake of an athlete is insufficient to meet the demands of both exercise and normal physiological function [2]. The term REDs stems from the earlier term “Female Athlete Triad”, initially defined in 1993 [3] as the interrelationship between disordered eating, amenorrhea, and osteoporosis. REDs recognises a broader range of health and performance consequences in both female and male athletes. This syndrome is associated with a wide array of clinical outcomes, including menstrual dysfunction, impaired bone health, compromised immune function, gastrointestinal disturbances, fatigue, reduced cardiovascular and metabolic function, and poor mental health [4,5]. Contributing factors to REDs are multifactorial and may include excessive training, inadequate nutrition, mental health challenges, disordered eating or eating disorders, poor sleep, illness, or undiagnosed medical conditions [2]. REDs is highly prevalent in elite sport settings; for instance, studies report signs of LEA in both male volleyball players and professional female football players, with many athletes at risk of developing the full spectrum of REDs [6,7]. Around 70% of elite athletes have a medium to high risk of REDs [8], but some studies have also shown a higher prevalence of symptoms. For instance, an Australian study found that 80% of elite and pre-elite athletes exhibited at least one REDs-related symptom, and 37% presented with two or more [9]. Prevalence is not only applicable in a sports setting but also in a clinical setting as Functional Hypothalamic Amenorrhea (FHA), which accounts for 20–35% of secondary amenorrhea cases [10].

The widespread prevalence and serious health implications of REDs underscore the importance of effective management strategies, both pharmacological and non-pharmacological, which have been identified in athletic populations [2] as such strategies help mitigate the impact of REDs on athletes [1]. Currently, non-pharmacological treatment forms the cornerstone of REDs management, with an emphasis on addressing the underlying issue of LEA. These strategies primarily involve educational initiatives aimed at increasing awareness and understanding of REDs among athletes, coaches, and healthcare professionals (a primary intervention strategy [11]). In addition, tertiary prevention strategies centered around individualized nutritional interventions are critical and show promising effects [11], focusing on increasing overall energy intake to restore energy balance and support physiological recovery [12]. Modifications to training load, by reducing volume or intensity, are also commonly implemented to decrease Energy Expenditure (EE) and allow for physiological restoration [13]. These non-pharmacological approaches are often effective in reversing early symptoms of REDs and remain the first-line treatment, particularly in the absence of severe clinical manifestations. Where non-pharmacological modalities of treatment are ineffective (notably in particular groups like females with resistant amenorrhea or with low bone mineral density (BMD)), not appropriate, or when physiological effects mean immediate intervention is required, pharmacological strategies can be considered [14]. Supportive treatments, such as calcium, vitamin D, and iron supplementation, are often implemented as part of a broader therapeutic strategy [11,15], although more specific strategies have been tested and utilised. For instance, transdermal 17β-oestradiol with cyclic progesterone has been shown to be more effective than combined oral contraceptives (COCs) in improving BMD in oligo-amenorrhoeic athletes [16]. Further to this, hormone replacement or bone-active agents may target complications, such as low bone density, when conservative measures fail [16,17]. However, these treatments can be considered controversial due to long-term safety concerns [18], and further research is required to clarify their role in REDs management.

Recent reviews, including those by Tenforde et al. [19] and Melin et al. [15], provide important insights into the pathophysiology and management of REDs. However, neither uses a systematic search process, with only one [19] undertaking a search of a database in 2014. Both reviews rely on a largely narrative summary of the results, not including quality assessment of research or certainty assessment. From these reviews, there is an evident lack of high-quality comparative evidence on treatment efficacy. The International Olympic Committee’s (IOC) narrative review on primary, secondary, and tertiary prevention strategies underscores the importance of addressing LEA as the root cause, recommending tailored nutritional rehabilitation, training adjustments, and multidisciplinary care [11]. Past review evidence and position statements consistently call for more evidence-based guidelines and emphasize that most pharmacological options address secondary outcomes (e.g., menstrual restoration or bone health) rather than the underlying LEA. To the best of the authors’ knowledge, no systematic review or meta-analysis has directly compared the efficacy, risks, and outcomes of pharmacological versus non-pharmacological treatments for REDs. This gap is critical given the rising prevalence of REDs and the growing diversity of affected athletic populations. Thus, the present systematic review and meta-analysis aims to fill this gap by rigorously evaluating available interventions for REDs.

## 2. Methods

A systematic review was undertaken and reported according to the PRISMA checklist [20]. A protocol was developed and published on 17 June 2025 on PROSPERO (reference: CRD420251073240).

### 2.1. Eligibility Criteria

Studies were included if they met the following conditions as set out by the Population, Intervention, Comparison, Outcome, and Study design (PICOS) acronym.

### 2.2. Participants

To be included, studies need to include female athletes classified as elite, recreational, pre-elite, club, active, or other levels. All athletes had to have characteristics identified within the remit of REDs [1]. Studies were included if they identified participants with oligo-amenorrhea, LEA, FHA, and/or exercise-related menstrual dysfunction (ExMD). Studies were excluded if participants were male, as limited research has been conducted on REDs in the male population. Studies were also excluded if participants had a clinical diagnosis of a psychological disorder or mental illness, were currently using hormonal contraceptives, or had Polycystic Ovary Syndrome (PCOS) or endometriosis. Studies using the same group of participants without evaluating different outcome measures were excluded to avoid duplication of results.

### 2.3. Intervention

Interventions were grouped as pharmacological or non-pharmacological interventions. Pharmacological interventions were included if they utilised Oestrogen therapy, Hormone Replacement Therapy (HRT), or calcium supplementation. Non-pharmacological interventions were included if they were based on diet alteration or dietary supplements, exercise, education, or consultation. Interventions were excluded if they did not directly target REDs, if their outcome measures were not relevant to REDs-related outcomes (see outcome eligibility criteria below), or if they focused exclusively on psychological measures (such as disordered eating, eating disorders, body image) without addressing the broader spectrum of REDs symptoms. This was identified to limit the clinical heterogeneity between different mental health diagnoses. It should be noted that any presence of a pharmacological intervention, e.g., (combined hormone therapy), when identified alongside a non-pharmacological intervention (e.g., vitamin D supplementation), was classified as pharmacological for the purpose of this review.

### 2.4. Outcomes

Studies were included if they utilised outcomes that included measures related to menstrual resumption, energy or dietary improvement related outcome measures, and physiological measures related to REDs, for instance, weight, fat mass, body percentage weight, cortisol, identification of hormones, and BMD. Alternative studies could be included if they identified physical or functional measures change (e.g., strength improvements) or psychological measures (e.g., mood, body image scale).

### 2.5. Study Design

Studies were included if they reported on the experience of, or outcomes from, an intervention-based study. Experimental, quasi-experimental, and pre-experimental designs were included. Conference proceedings, theses, and ongoing research were included to reduce the risk of publication bias. Studies were excluded if they did not report on the experience of, or outcomes from, an intervention-based study. Non-experimental designs, such as cross-sectional research pieces, were excluded. Reviews, editorials, opinion pieces, and commentaries were also excluded. Studies that were unpublished without accessible data, duplicate publications, and conference abstracts without full data were excluded unless further data were available.

### 2.6. Other Criteria

Date restrictions were not applied to the search dates. No restriction on language was made. No publications written in languages other than English were used. Where any translation could not be comprehended, the study was excluded.

### 2.7. Information Sources and Search Strategy

A blind search by two authors (A.W., A.S.), supported by the management software Covidence 2025 ©, was undertaken from a total of five electronic databases. They included CINAHL, MEDLINE, SportDiscus, ERIC, and Embase from inception until July 2025. Search strategies combined controlled vocabulary (e.g., MeSH terms) and free-text keywords relating to the population, including women or females. These conditions include Relative Energy Deficiency in Sport (REDs), female athlete triad, low energy deficiency, amenorrhea, oligomenorrhea, and menstrual disturbance. The intervention includes terms like intervention, treatment, therapy, female, or women. The search was adapted for each database, with Boolean operators, truncations, and limits applied as appropriate. Full search strings for each database are provided in Appendix A. In addition to this, three electronic search engines, Google Scholar, ScienceDirect, and Findit.Bham, were searched for the first 30 pages of results using the terms “females and relative energy deficiency and sport”. The gray literature was searched using the Grey Matters search engine. Citation chasing was undertaken with all included articles and all identified previous reviews.

### 2.8. Selection Process

Two blind authors (A.W., A.S.) undertook the selection process using Covidence 2025 © software. A separate academic with experience in systematic reviews was available to arbitrate discussions when a decision could not be made (two studies were identified for this process, which were included following arbitration). Both authors made decisions by title, then by abstract, and then by reading the full text independently.

### 2.9. Data Items

A pre-determined extraction data tool devised in Microsoft Excel identified specific variables that included demographic variables (study title, journal, design, type of intervention, geographical location, population, age, sport, athlete level, participant identifying group, if a clinical diagnosis was obtained, and method of assessment). The extraction tool was pilot tested on three studies and refined before full extraction occurred. Intervention variables were identified according to the TEDieR guidelines [21], tabulated, and outcome measures (identification of all outcomes and identification of primary and secondary outcome measures) were considered, which were then grouped by domains.

### 2.10. Risk of Bias Assessment

A risk of bias assessment was undertaken using the ROB-2 tool [22] for randomised control trials or the Robins-I Version 2 tool [23] for non-randomised control trials. For other types of design or instance case control or case series, the JBI critical appraisal tools were utilised (https://jbi.global/critical-appraisal-tools, accessed on 25 June 2025). See the Appendix A for the assessment.

### 2.11. Effect Measures

Mean differences were considered where possible for all outcome measures. Fixed effects models were used for the meta-analysis. Statistical heterogeneity was quantified using I^2^. Thresholds of 25%, 50%, and 75% represented low, moderate, and high heterogeneity [24]. However, we did not use these as absolute cut-offs and did consider clinical and methodological diversity of studies [25]. When the standard deviation of change scores could not be identified, a formula for estimating it was used as follows:$$\mathrm{Standard}\;\mathrm{deviation}\;(\mathrm{SD})\;\mathrm{of}\;\mathrm{change}\;=\;\surd ({\mathrm{SD}}^{\mathrm{baseline}2}\;+\;{\mathrm{SD}}^{\mathrm{final}2}\;-\;2\cdot r\cdot {\mathrm{SD}}^{\mathrm{baseline}}\cdot {\mathrm{SD}}^{\mathrm{final}})$$

No multi-arm studies were used in the meta-analysis conducted. All meta-analyses were conducted using RevMan version 5.4.

### 2.12. Synthesis

A narrative synthesis documented the results by outcome domain. The following outcome domain areas were identified (a) physiological (including menses and menses restoration, weight, cortisol, identification of hormones, BMD), (b) physical and functional (including measures of strengths or function), and (c) psychological outcomes (including mood, body image scale, or eating disorder questionnaire). A meta-analysis was possible and conducted for three outcome measures as part of the synthesis for non-pharmacological interventions. Requirements for the meta-analysis included having at least 3 studies that used consistent measurement methods, outcome measures that were comparable, and intervention durations that were consistent, ensuring that clinical characteristics of the included studies were homogenous. A primary focus was on physiological outcome measures, which were most consistently reported. Clinical heterogeneity, heterogeneity in measurement tools, and insufficient data meant that no other meta-analysis was possible due to the limited evidence currently available.

### 2.13. Certainty Assessment

GRADE [26] was used to assess the certainty of the outcome measure results and supplement narrative synthesis and meta-analysis.

### 2.14. Equity, Diversity, and Inclusion Statement

Authors used a narrative synthesis to identify common demographics and intervention characteristics. The group selected within the eligibility criteria and represented across studies are reasonably homogenous due to the individuals representing studies mainly from the USA and from university-based settings that represent female endurance and distance sports/athletes. The focus of this work was only on females and is not extended to males.

### 2.15. Patient and Public Involvement

No patient or public involvement in this review was undertaken.

## 3. Results

### 3.1. Study Selection

A total of 3156 articles were examined by both reviewers. A total of 261 were input into the Covidence database, of which 162 were identified for screening, and 19 studies met the inclusion criteria for this systematic review. This comprised fifteen [27,28,29,30,31,32,33,34,35,36,37,38,39,40,41] (15/19, 79%) non-pharmacological interventions and four [16,42,43,44] (4/19, 21%) pharmacological interventions. See Figure 1 for the PRISMA flow diagram (see Appendix A for additional information).

### 3.2. Study Characteristics

A total of 759 female participants were included (n = 759). This included individuals most often in the age bracket of 18–35 years. Where possible (n = 12 studies, n = 473 participants), an aggregated mean age was calculated as 20.9 years. The most included group of females was female ‘athletes” (n = 12/19, 63%), followed by “exercising” or “active” females (n = 5/19, 26%). Non-pharmacological trials included nine (9/15, 60%) as athletes, five as “exercising” or “active” females (5/15, 33.33%), and one (1/15, 7%) identified without classification. Pharmacological trials included athletes (3/4, 75%), and one study included ballet dancers (1/4, 25%). The classification of included sports most often related to a multi-sport, 7/19 (37%; 6/15, 40% for non-pharmacological and 1/4, 25% pharmacological) or a multi-sport with the term endurance or distance, 10/19 (53%; 8/15, 53% for non-pharmacological and 2/4, 50% for pharmacological). The most common specific sport mentioned was running, followed by cycling. Although, exact numbers for these specific sports would be hard to determine due to the use and inclusion of multi-sports. The most common country location for studies was the USA (n = 12/19, 60%), including 10/15 (67%) for non-pharmacological interventions and 2/4 (50%) for pharmacological; this was followed by Germany (n = 3/19, 16%), all coming from non-pharmacological interventions. See Table 1 for a summary of non-pharmacological interventions and Table 2 for a summary of pharmacological interventions.

### 3.3. Intervention Characteristics

The most common characteristics across studies was as follows. The classification system used to identify athletes most often identified amenorrhea (n = 6/19) as a clinical inclusion criterion for participation. Common intervention components included (a) increasing energy intake/diet supplementation (n = 10/19; 53%; 10/15, 67% for non-pharmacological interventions) and (b) dietary/nutritional counseling or advice (out of all studies, n = 6/19, 32%), and 6/15 (40%) for non-pharmacological interventions was the most prevalent intervention component. The average duration of intervention for non-pharmacological interventions was 28.3 ± 15.8 weeks (n = 12/15, 80%). The setting selected for the intervention was most often described as a “university” setting (n = 8/19, 42%), which included 8/15 (53%) for non-pharmacological studies. Staff involved in the interventions were most often identified as dietitians (n = 12/19), 12/15 (80%) for non-pharmacological studies and 1/4 (25%) for pharmacological interventions, or as researchers (n = 9/19, 47%), including 8/15 (53%) for non-pharmacological interventions and 1/4 (25%) for pharmacological evidence. See Table 3 for a TIDieR summary of the non-pharmacological studies and Table 4 for a TIDieR summary table for the intervention studies.

### 3.4. Critical Appraisal

The following risk of bias tools were used depending on study design: ROB 2 (for Randomised control trials) (RCTs, n = 7/19, 37%), ROBINS-I V2 (for non-randomised interventional studies, n = 11, 58%), and the JBI checklist (for case reports, n = 1, 5%).

Overall, RCTs demonstrated low (n = 3/7, 43%) to moderate (n = 4/7, 57%) risk of bias, with most (n = 5/7, 71%) studies showing low risk across all key domains. However, two RCTs [40,42] had “some concerns”, particularly in randomisation and outcome reporting. In contrast, all non-randomised studies were rated as having serious risk due to the influence of confounding variables (e.g., baseline differences of energy availability, variation in training load, duration of menstrual dysfunction, age). Additional concerns were noted in non-RCTs in domains, such as intervention classification (serious risk (n = 1/11, 9%), moderate risk (n = 10/11, 91%)), outcome measurement (n = 10/11, 91%), and missing data (n = 6/11, 55%), typically rated at moderate risk. The single case report by Mallinson et al. [38] was methodologically sound across most JBI criteria but lacked reporting on adverse or unanticipated events. Appendix A provides further information regarding critical appraisal.

### 3.5. Synthesis and Certainty Assessment

Findings are provided below for non-pharmacological evidence and pharmacological evidence. Appendix A is available that provides the assessment of critical appraisal (quality) and certainty assessment.

### 3.6. Non-Pharmacological

Among the studies assessing non-pharmacological interventions, the most frequently reported outcomes were menstrual function recovery (n = 8/15, 53%), improvements in energy availability (EA) (n = 5/15, 33%), changes in body composition (n = 7/15, 47%), and alterations in relevant biomarkers (n = 9/15, 60%). These are reported below.

### 3.7. Menstrual Function Recovery

Menstrual function recovery was utilized as an outcome measure in eight (n = 8/15, 53%) of the included papers. Seven (n = 7/8, 88%) reported statistically significant improvements (*p*-values ranging from <0.01 to 0.05), with large, estimated effect sizes (0.8–1.2). One case report lacked statistical analysis but described clinically meaningful recovery in both participants. All nine studies were judged to show clinically meaningful outcomes, primarily associated with increased EA through nutritional (n = 5/8, 63%) or non-pharmacological interventions (n = 1/8, 13%) or both nutritional and non-pharmacological interventions (n = 2/8, 25%). Six (6/8, 75%) studies reported the number of intervention participants who experienced improved or recovery of menses. Of the 142 intervention group participants, 47 (47/142, 33%) had partial or full recovery of menses.

### 3.8. Confidence in Evidence

Summary of evidence contributing to certainty rating: GRADE study ratings included one very low, five low, and two high.

The overall confidence rating was low, although findings can be taken with reasonable confidence due to the consistency of evidence.

### 3.9. Energy Availability

Changes in EA were evaluated in five studies (n = 5/15, 33%). All five reported statistically significant improvements (*p*-values ranging from <0.01 to 0.05), with estimated effect sizes between 0.5 and 1.0. Each study demonstrated clinically meaningful improvements, achieved through increased caloric intake (n = 5/15, 33%).

### 3.10. Confidence in Evidence

Summary of evidence contributing to certainty rating: GRADE study ratings included five low.

The overall confidence rating was low, although the findings can be taken with reasonable confidence due to the consistency of evidence.

### 3.11. Body Composition Measures

Body composition was measured in seven (n = 7/15, 47%) of the non-pharmacological papers. Six (n = 6/7, 86%) reported statistically significant improvements in at least one parameter, such as body weight, fat mass, or body fat percentage, with *p*-values ranging from <0.001 to 0.05 and effect sizes between 0.5 and 2.5. One study did not report statistical significance but showed large, estimated effects. All seven studies were considered clinically meaningful, with improvements primarily associated with increased energy intake. Meta-analysis was undertaken for the measures of changes in fat mass and percentage of body fat (see Figure 2 and Figure 3). The change in body mass (kg) was identified as 1.36 (95% CI 0.68, 2.04). Statistical heterogeneity was identified as low (I^2^ = 0%). The change in body fat percentage (%) was identified as 2.21 (95% CI 1.34, 3.08). Statistical heterogeneity was identified as moderate (I^2^ = 32%).

### 3.12. Confidence in Evidence

Summary of evidence contributing to certainty rating: GRADE study ratings included five low and two very high.

The overall confidence rating: moderate, although findings can be taken with reasonable confidence as the evidence is supported by a meta-analysis.

### 3.13. Hormonal and Blood Biomarkers

Biomarkers related to REDs were analysed in nine (n = 9/15, 60%) studies. Eight (n = 8/9, 89%) reported statistically significant improvements (*p* < 0.05), with effect sizes ranging from 0.5 to 0.8. The most assessed markers included serum leptin, triiodothyronine (T3), cortisol, oestradiol, and luteinizing hormone (LH) pulsatility. One study, while lacking statistical testing, demonstrated substantial changes in multiple biomarkers and was considered clinically meaningful. All nine studies reported clinically meaningful outcomes, supporting the utility of biomarker monitoring in evaluating recovery in REDs and the effectiveness of dietary and non-pharmacological interventions. A meta-analysis (see Figure 4) was possible regarding the reporting of T3 by three studies. The results identified a mean change of −2.37 (95% CI −5.57, 0.83) on T3 biomarkers (nmol/L). Statistical heterogeneity was identified as high (I^2^ = 91%).

### 3.14. Confidence in Evidence

Summary of evidence contributing to certainty rating: GRADE study ratings included six low, three very high, and eight with a moderate effect.

The overall confidence rating was low; evidence was affected by quality downgrades, and a meta-analysis favored the control group. Further evidence is required.

### 3.15. Pharmacological

Studies examining pharmacological treatments commonly assessed BMD (n = 4/4, 100%), hormonal profiles (e.g., oestrogen, L, follicle-stimulating hormone (FSH)) (n = 3/4, 75%), recovery of menstrual function (n = 3/4, 75%), and bone turnover markers (n = 1/4, 25%) as indicators of skeletal health and remodeling.

### 3.16. Bone Mineral Density Markers

BMD was assessed in four (n = 4/4, 100%) pharmacological studies.

Ackerman et al. [16] identified only the protective effect on BMD of one intervention group named PATCH (physiological oestrogen replacement via 100 mcg transdermal 17β-E2 and 200 mg cyclic micronized progesterone) versus no intervention. They identified protective effects as risk ratios of 11.22 (95% CI; 2.12–59.29; *p* = 0.008) for the spine, 10.76 (95% CI; 2.07–55.98; *p* = 0.001) for the femoral neck, and 1.44 (95% CI; 0.19–10.76; *p* = 0.292) for the total hip. The results were controlled for age, height, race, ethnicity, and weight change. The PILL (combined oral contraceptives of 30 µg ethinyl estradiol with 0.15 mg desogestrel) identified no differences.

Dadgostar et al. [42] utilized a low dose oral contraceptive (30 µg ethinyl estradiol and 150 µg levonorgestrel) and identified no statistically significant changes in a small sample (n = 8 for the intervention group and n = 5 for control group that received calcium and vitamin d) but identified a slight increase (0.3% increase) in BMD values at the lumbar spine at 9 months in those receiving and a minor decrease (1.6%) in the control group over the same time period. The effect size when comparing the change scores was *d* = 0.23 for the spine (small effect) compared to the control and *d* = 0.11 for the femur (limited or no effect) compared to the control.

Gibson et al. [43] utilized three groups, the intervention group with 1 mg estriol and 2 mg estradiol for 12 days and then 1 mg estriol, and 1 mg of estradiol for 6 days, and 1000 mg calcium carbonate (n = 10) versus the calcium carbonate (1000 mg; n = 14) group versus the control group with no treatment (n = 10). The intervention group was identified as having positive changes in the following locations expressed by percentage change: the trochanteric region (1.71%, SD = 2.86), the lumbar spine L2-L4 (5.67%) (SD = 9.47), and Ward’s triangle (3.55%) (SD = 4.05). This resulted in effect sizes compared to the control group of *d* = 0.73 for the trochanteric region, *d* = 0.80 for the lumbar spine, and *d* = 0.94 for Ward’s triangle. The neck of the femur showed a decrease of −0.63% (SD = 2.21), although this still created a small positive effect size of *d* = 0.18 against the control group.

The calcium group identified positive changes but high standard deviations in the neck of the femur (1.33%) (SD = 6.29) and Ward’s triangle (1.33%) (SD = 9.00). The effect size compared to the control group was a small effect size of *d* = 0.26 at the neck of the femur and *d* = 0.29 at Ward’s triangle. The other sites, including the trochanteric region (−0.33%, SD = 5.21) and the lumbar region (−0.03, SD = 5.01), demonstrated a small negative change.

Warren et al. [44] compared the intervention group (n = 13) with Premarin (0.625 mg; 25 days) and then with Provera (10 mg, 9 days [days 16–25]) versus placebo (n = 11). The intervention group identified some positive percent change in BMD across the 24-month period at the spine (5.60) (SD = 1.10) and wrist (0.91) (SD = 5.12) but not at the foot (−6.49) (SD = 2.04). The placebo group arguably outperformed the intervention group with a positive change across all sites, including the spine (4.46%) (SD = 2.80), the wrist (3.19%) (SD = 1.48), and the foot (1.48%) (SD = 2.83).

### 3.17. Confidence in Evidence

Summary of evidence contributing to certainty rating: GRADE study ratings included one very low, one low, one high, and one very high.

The overall confidence rating was moderate, and the findings can be taken with reasonable confidence due to the consistency of evidence, but some caution is needed when considering the exact pharmacological intervention used.

### 3.18. Hormonal Profiling

Hormonal profiles were evaluated in three (n = 3/4, 75%) studies, with two [16,43] (n = 2/4, 50%) showing significant improvements (*p* < 0.05) in key hormones, such as oestradiol and progesterone, with effect sizes around 0.7–0.8. One study [44] reported normalization of hormonal levels in treatment groups with significance at baseline. Three [16,43,44] (n = 3/4, 75%) studies assessing hormonal outcomes were considered clinically meaningful.

Ackerman et al. [16] identified a significant number of results comparing two intervention groups against the control. See Table 5 for a summary of the effect size identified comparing all groups at 6 months and 12 months.

Dadgostar et al. [42] only measured lipid profiles and apolipoprotein levels and did not report hormonal data, and Warren et al. [44] only identified hormonal data at baseline.

### 3.19. Confidence in Evidence

Summary of evidence contributing to certainty rating: GRADE study ratings included one very low, one low, and one very high.

The overall confidence rating was low. The findings can be taken with caution due to inconsistency of reported outcomes and evidence.

### 3.20. Menstrual Function Recovery

Menstrual function recovery was reported in three (n = 3/4, 75%) [16,43,44] studies, all demonstrating statistically significant improvements (*p*-values ranging from *p* < 0.05 [43] to 0.0001 [16]). These improvements were deemed clinically meaningful, indicating effective restoration of menstrual function with pharmacological treatment.

### 3.21. Confidence in Evidence

Summary of evidence contributing to certainty rating: GRADE study ratings included one very low, one low, one high, and one with a large effect and a dose response.

The overall confidence rating was moderate, and the findings can be taken with reasonable confidence due to the consistency of evidence.

### 3.22. Bone Turnover Markers

Bone turnover markers were evaluated in one (n = 1/4, 25%) study [16], which found significant changes in markers, including P1NP and IGF-1 (*p* = 0.016), with clinically meaningful effects observed. Effect sizes can be observed in Table 6. Changes in P1NP over 12 months were positively associated with changes in estradiol (r = 0.35, *p* = 0.004) and IGF-1 (r = 0.37, *p* = 0.003) and inversely with changes in SHBG (r = −0.28, *p* = 0.019). For changes in BMD over time, changes in estradiol were associated with changes in the femoral, neck, spine, and hip BMD at 12 months (r ≥ 0.27, *p* ≤ 0.024).

### 3.23. Confidence in Evidence

Summary of evidence contributing to certainty rating: GRADE study ratings included one study rated as high certainty.

The overall confidence rating was low. Further evidence is required to repeat the results.

### 3.24. Comparative Overview of Intervention Effects

Across the included studies, non-pharmacological interventions consistently demonstrated improvements in menstrual recovery, energy availability, and body composition, with meta-analyses confirming modest but significant benefits for fat mass and body fat percentage. However, certainty of evidence was generally rated low to moderate, reflecting methodological limitations and heterogeneity in outcome definitions. Pharmacological interventions yielded some (at times) very positive changes in hormonal and bone-related biomarkers, often with clearer short-term physiological effects, but the small number of trials and sample sizes utilized and concerns about masking underlying energy deficiency limited confidence in their broader applicability. Taken together, these patterns suggest that while both approaches can produce measurable benefits, current evidence included here suggests that non-pharmacological strategies offer more consistent though less certain improvements across multiple domains, whereas pharmacological options provide targeted effects supported by fewer, but sometimes higher-certainty, studies.

## 4. Discussion

This review provides initial insight and certainty of evidence considering the effectiveness of non-pharmacological interventions compared to pharmacological interventions. Non-pharmacological interventions demonstrated consistent improvements across multiple outcome domains, including recovery of menstrual function, EA, body composition, and hormonal biomarkers. Pharmacological interventions, utilising oestrogen therapy and hormone replacement, showed some effectiveness in improving BMD, restoring menstrual function, and enhancing hormonal profiles. However, the certainty of evidence across both intervention types was limited by methodological weakness and the number of contributing studies. The discussion now provides consideration of the main findings by intervention type.

### 4.1. Non-Pharmacological Interventions

#### 4.1.1. Menstrual Function Recovery

Despite the overall low quality of evidence, all studies consistently demonstrated clinically meaningful improvements in menstrual function following non-pharmacological interventions. These findings align with the broader physiological and clinical literature. Mechanistically, energy deficiency resulting from insufficient fuel availability suppresses the hypothalamic–pituitary–ovarian (HPO) axis, leading to menstrual dysfunction [45]. Supporting this, research shows that LH pulsatility is disrupted when EA drops below a clinical threshold, highlighting that EA, rather than body fat or exercise alone, is the primary regulator of reproductive function in active women [46]. Clinical evidence further supports this, with several studies showing that dietary or training modifications can restore menses within several months in affected athletes [27,35].

Current studies vary widely in how recovery is defined, often relying on menstrual bleeding alone [27,28,35,36], which does not confirm ovulation or hormonal restoration [34,38]. Menstrual bleeding alone does not guarantee hormonal balance or ovulation, highlighting the need for standardized, hormonally validated definitions in future research. Future research should adopt standardized, hormonally validated definitions of menstrual recovery to improve consistency and clinical relevance. In addition, longer follow-up periods and consistent use of hormonal markers are needed to assess the long-term health impacts of interventions on reproductive, bone, and overall health in female athletes with REDs.

#### 4.1.2. Energy Availability

EA was identified as improving following interventions. These findings are strongly supported by past research demonstrating the central role of EA in hormonal, reproductive, and metabolic regulation. Low EA has been shown to impair LH pulsatility, suppress resting metabolic rate, and reduce estrogen and IGF-1 levels, negatively affecting reproductive and bone health [47,48]. Importantly, even modest increases in EA can reverse these changes, restoring LH pulsatility, resuming menstrual function, and normalizing metabolic function, highlighting EA as a key modifiable factor in both the development and recovery from REDs. Improvements in bone health have also been observed with prolonged energy restoration, though severe cases may require combined nutritional and pharmacological intervention [49].

Accurate measurement of EA remains a major challenge due to the difficulty of precisely assessing dietary intake, exercise, and fat-free mass. Common reliance on self-reported data and indirect proxy markers, such as menstrual dysfunction, resting metabolic rate, and hormonal changes (reduced leptin and T3), introduces variability and error, limiting confidence in the current findings [17,47]. Therefore, future research needs improved, objective methods to accurately quantify EA in free-living athletes, alongside standardized and sensitive biomarkers, to strengthen the evidence base and guide effective REDs management.

#### 4.1.3. Body Composition

Consistent and clinically relevant improvements in body composition were observed across studies.

Past evidence highlights the complexity of using body composition as a primary indicator of recovery in REDs. Although improvements in fat mass and body weight may reflect enhanced EA and nutritional rehabilitation [17], they are not universally required for physiological recovery. Critically, key outcomes such as the return of menses and improvements in BMD, central to REDs recovery, can occur independently of significant changes in body composition [50]. This is particularly relevant in athletes who are constitutionally lean, where minimal or no changes in fat mass may accompany full recovery of hormonal and metabolic function [51]. Moreover, reliance on body composition alone may overlook meaningful clinical progress or delay appropriate intervention if weight change is minimal. As such, the current consensus, including the 2023 IOC statement on REDs [1], emphasizes that body composition should not be used in isolation to define recovery. Instead, it should be interpreted within a broader clinical framework, incorporating menstrual status, hormonal markers, non-pharmacological changes, and performance indicators. This multidimensional approach is essential for accurately monitoring recovery and tailoring individualized treatment strategies in athletes affected by REDs.

#### 4.1.4. Biomarkers

Endocrine markers, particularly serum leptin and triiodothyronine (T3), along with LH and estradiol, showed consistent, clinically meaningful improvements across studies, indicating recovery of metabolic and reproductive function in REDs. The current findings are consistent with the previous literature highlighting the utility of endocrine biomarkers as sensitive indicators of physiological recovery in REDs. Increases in serum leptin, an adipocyte-derived hormone that signals energy sufficiency to the hypothalamus, are closely associated with improvements in reproductive function [52,53]. Similarly, elevations in T3, a well-established marker of metabolic adaptation, reflect the reversal of energy-conserving mechanisms and the restoration of metabolic homeostasis [17]. Additional improvements in cortisol, estradiol, and LH pulsatility further support the reactivation of the hypothalamic–pituitary–gonadal (HPG) axis [19,47]. Given the low certainty of current evidence, future research must focus on high-quality, standardized studies to validate endocrine biomarkers as reliable indicators of REDs recovery. Markers like leptin, T3, estradiol, and LH pulsatility show promise but are limited by inconsistent measurement, variability, and unclear clinical thresholds [17,47]. To improve utility, studies should standardize biomarker timing, consider menstrual cycle, diurnal variation, and assay methods, and establish validated cut-offs. Longitudinal research linking hormonal changes to clinical outcomes, such as menstrual resumption and bone health, is needed. Importantly, biomarkers should be combined with clinical assessments, symptom tracking, and performance measures for a comprehensive evaluation [16]. Rigorous validation of these markers is essential to enhance REDs diagnosis, monitoring, and management.

#### 4.1.5. Pharmacological Interventions

##### Bone Mineral Density

BMD outcomes in REDs interventions were inconsistent across studies, with some evidence suggesting that hormonal therapies may support bone health when nutritional recovery alone is inadequate.

These findings align with existing evidence suggesting that while hormonal therapies may offer some benefit to BMD [50], their effectiveness remains inconsistent. For instance, a review by Indirli et al. [54] reported that hormone therapies using estrogen or leptin showed limited impact on bone metabolism in women with FHA. This variability is likely since such treatments do not address the underlying cause of REDs: chronic low EA. Pharmacological interventions may alleviate certain symptoms but risk masking the broader physiological dysfunction if used in isolation [17,53]. As such, current evidence supports their use only as adjuncts in cases where nutritional rehabilitation and restoration of EA, the foundation of REDs treatment, have not been sufficient, or where bone health is severely compromised [55]. Further to this, out of the four studies currently identified, it should be noted that the strongest evidence was linked to one trial [16], with some support from another trial [43]. Understanding the interventions in these two studies would be important if considering pharmacological treatment as an adjunct.

##### Menstrual Function Recovery

Although pharmacological interventions have shown promise in supporting menstrual recovery in REDs, current evidence, particularly from non-pharmacological studies, suggests that nutritional and non-pharmacological strategies may offer equally, if not more, effective outcomes, albeit from a smaller and methodologically limited evidence base.

Evidence supporting the current findings aligns with established physiological mechanisms, indicating that hormonal therapy in REDs may obscure true recovery. Combined oral contraceptives (COCs) and other exogenous hormone regimens induce withdrawal bleeding through artificial endometrial shedding, without restoring endogenous ovulatory cycles or HPG axis function [56,57]. As such, withdrawal bleeding can be misinterpreted as menstrual recovery, potentially delaying appropriate treatment interventions targeting LEA. This masking effect is well-documented and underscores the need for more accurate markers of reproductive recovery. Objective indicators such as serum progesterone levels, LH pulsatility, or basal body temperature tracking are recommended to assess ovulatory function and HPG axis restoration [58]. Given these considerations, hormonal therapies should be used with caution and only as adjuncts to primary nutritional and non-pharmacological strategies that address the underlying energy deficiency central to REDs.

##### Hormonal Profiles

Hormonal interventions demonstrated significant and clinically meaningful effects on estrogen and progesterone levels, with estradiol notably higher in treatment groups compared to those receiving oral contraceptives. But, the certainty of evidence is mixed.

Ackerman et al. [16] demonstrated that improvements in BMD can occur without endogenous hormonal normalization, as menstrual function did not resume during treatment with transdermal estrogen. This indicates that symptom recovery, such as bone health improvement, may happen independently of menstrual and hormonal recovery when exogenous hormones are administered. Interpreting endocrine responses in this context is complex because exogenous hormones, like transdermal estradiol or COCs, suppress endogenous hormone production via negative feedback on the HPG axis [16,59]. This suppression can create misleading hormonal profiles that suggest normalization without true recovery of natural reproductive function, such as ovulation and menstrual cycles [59]. Therefore, clinical improvements in outcomes may reflect the direct effects of hormone therapy rather than restoration of endogenous endocrine function [16]. These complexities highlight the need for a comprehensive assessment of REDs recovery that goes beyond hormone levels to include clinical symptoms and functional reproductive status [17].

##### Bone Turnover Markers

Analysis of bone turnover markers, specifically P1NP and IGF-1, showed statistically and clinically significant changes indicative of meaningful effects on bone metabolism, supported by high-certainty evidence; however, the findings are limited by being based on a single study, highlighting the need for further research to validate their utility in monitoring bone health in athletes with REDs.

These findings are supported by the literature recognizing bone turnover markers, such as P1NP and IGF-1, as sensitive indicators of bone metabolism [60]. However, their clinical utility is challenged by variability in assay methods, biological fluctuations, and influences from nutrition, exercise, and hormonal status [61]. The limited number of studies reporting these markers, alongside inconsistent measurement protocols and definitions, further complicates cross-study comparisons [62]. Nevertheless, bone turnover markers provide valuable early insight into bone remodeling processes that may occur before detectable changes in BMD, emphasizing the importance of standardized assessment methods and additional research to fully establish their role in monitoring bone health in REDs.

#### 4.1.6. Limitations of the Evidence

Most included studies forming the evidence were rated as having moderate to high risk of bias, and the consistency of using a standard set of reporting outcome measures was poor. This limits the strength of the evidence base. Reporting on intervention fidelity, participant compliance, and follow-up was often incomplete or unclear, reducing confidence in the consistency and applicability of reported outcomes. These methodological limitations highlight the need for more robust, standardized research in this field. Another limitation is that the standard deviation of change used in the meta-analysis for the Dueck [31] was estimated. It is important to recognize characteristics of the studies that limit the results. These include the variability in diagnostic criteria for REDs and the other conditions identified, inconsistency in definitions of menstrual recovery (e.g., bleeding versus ovulation), and the impact of access to cohorts, which were often based at a university from high-income countries and not fully powered and, in some cases, had very small sample sizes. Where interventions combined pharmacological with non-pharmacological components, the impact of each was not assessed. Finally, REDs is acknowledged as affecting both genders, and this review is limited by a focus on females.

#### 4.1.7. Implications for Practice/Clinical Implications

##### Implications for Practice (Expanded)

The current review primarily included female athletes engaged in distance and endurance sports, with most non-pharmacological interventions focusing on dietary modifications delivered in university-based settings. Effective management of REDs should begin with restoring energy availability through nutritional strategies, as these remain the cornerstone of treatment and are consistently supported by evidence. Clinicians should avoid relying solely on menstrual status or body composition as indicators of recovery; instead, a multidimensional approach is recommended, incorporating hormonally validated markers such as leptin, T3, LH, and estradiol alongside behavioral changes to ensure accurate diagnosis, monitoring, and long-term management.

Hormonal therapies may offer benefits for bone health and symptom relief when nutritional rehabilitation and energy restoration prove insufficient; however, they do not address the underlying cause and can mask true reproductive recovery, warranting cautious use. Combined oral contraceptives, in particular, may induce withdrawal bleeding without restoring ovulatory function, underscoring the need to prioritize non-pharmacological interventions and objective monitoring. Finally, bone turnover markers, such as P1NP and IGF-1, show promise as sensitive indicators of bone metabolism, but their clinical utility remains limited by variability and insufficient evidence, highlighting the need for standardized assessment and further research.

##### Practical Implementation for Clinicians

In practice, clinicians should start by conducting a comprehensive assessment of energy availability, dietary intake, and training load, ideally in collaboration with a sports dietitian. Establishing realistic nutritional goals and monitoring adherence through food diaries or digital tracking tools can help ensure progress. Hormonal profiles and bone markers should be integrated into routine evaluations, using consistent timing and validated assays to improve reliability. When considering hormonal therapy, clinicians should clearly communicate its role as an adjunct rather than a primary treatment and set expectations regarding its limitations in restoring ovulatory function. Regular multidisciplinary reviews involving dietitians, psychologists, and sports physicians can support behavioral change and optimize long-term outcomes.

##### Implications for Research Design

Future research should move beyond small, single-center studies and prioritize larger, multi-center trials to improve generalizability and statistical power. Inclusion of male athletes is essential to address current gender gaps and broaden the applicability of the findings. Standardization of REDs diagnostic criteria and core outcome sets—including hormonal markers, bone health indicators, and validated measures of energy availability—would enhance comparability across studies. Longer follow-up periods are needed to capture sustained changes in bone mineral density and reproductive function, as short-term interventions may not reflect long-term recovery. Additionally, integrating behavioral and psychological outcomes alongside physiological markers could provide a more comprehensive understanding of treatment efficacy.

## 5. Conclusions

Non-pharmacological interventions, predominantly dietary energy restoration and adjustments to training load, consistently demonstrate benefits on clinically relevant outcomes in athletes with REDs. However, the certainty of this evidence is often low to moderate due to methodological limitations such as small sample sizes, short follow-up periods, and heterogeneity in outcome measures. In contrast, pharmacological interventions, including hormonal therapies, appear to improve bone mineral density and certain hormonal parameters, yet they do not address the underlying issue of low energy availability and may obscure true reproductive recovery. These findings underscore the importance of prioritizing nutritional and behavioral strategies as first-line management while reserving pharmacological approaches for carefully selected cases and ensuring ongoing monitoring of physiological recovery. Further research is needed.

## Figures and Tables

**Figure 1 sports-13-00453-f001:**
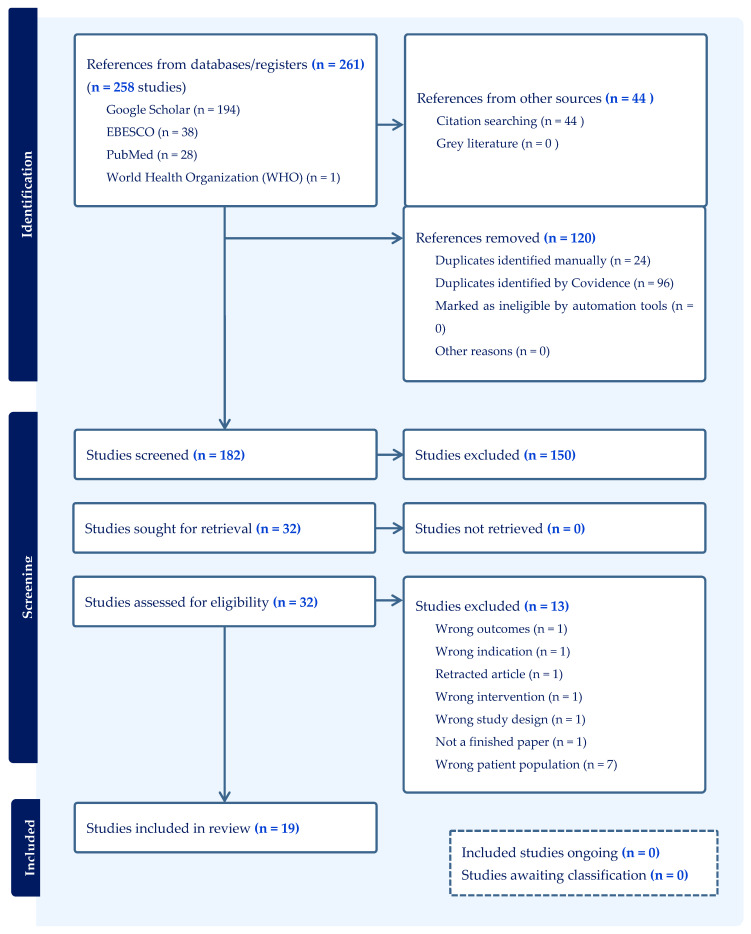
A PRISMA 2020 flow diagram produced by Covidence to identify the blind search process undertaken by two authors.

**Figure 2 sports-13-00453-f002:**
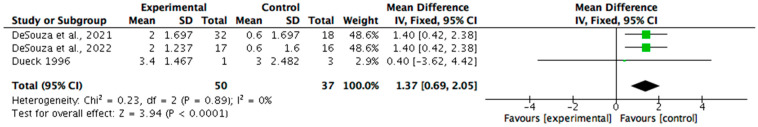
A meta-analysis showing the benefit of nutrition-based interventions on fat mass [29,30,31].

**Figure 3 sports-13-00453-f003:**
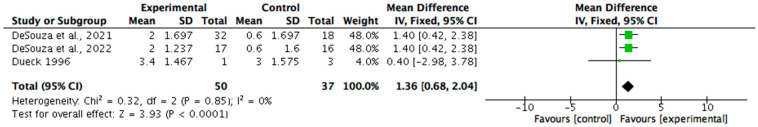
A forest plot showing the benefit of nutrition-based interventions on percentage body fat [29,30,31].

**Figure 4 sports-13-00453-f004:**
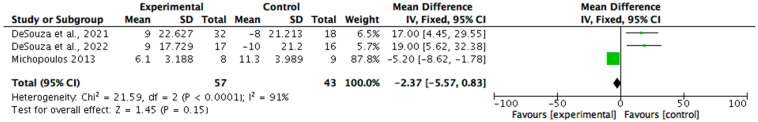
A forest plot showing the T3 biomarker changes from non-pharmacological interventions [29,30,40].

**Table 1 sports-13-00453-t001:** Summary characteristics of non-pharmacological interventions.

Authors	Study Design and Classification	Sample and Sample Size	Sport, Athlete Level, and Geographical Location	Participant Identification	Comparator or Control	Intervention Duration (Weeks)	Outcome Measures	Results
Arends et al. [27] (2012)	Design: Five year retrospective cohort study. Classification: Non-pharmacological	Female collegiate athletes (n = 51) (36 oligomenorrheic, 15 amenorrheic). Age range: 18–21 years	Sport: Multi-sport. Athlete level: Collegiate. Country: USA	Menstrual disturbances (oligomenorrhea, amenorrhea)	Within-group comparison: athletes who did vs. did not recover menses	260	Primary outcome: Time to ROM. Secondary outcomes: weight, BMI changes	17.6% (9/51) of athletes regained menses during follow-up; mean time to ROM = 15.6 ± 2.6 months; ROM group had significantly greater weight gain (5.3 ± 1.1 vs. 1.3 ± 1.1 kg), % weight gain (9.3 ± 1.9% vs. 2.3 ± 1.9%), and BMI increase (1.9 ± 0.4 vs. 0.5 ± 0.4 kg/m^2^) compared to non-ROM athletes; weight gain predicted ROM (OR 1.25, 95% CI 1.01–1.56)
Cialdella-Kam et al. [28] (2014)	Design: Intervention study. Classification: Non-pharmacological (carbohydrate–protein supplement).	Female athletes (n = 8). Age range: 19–27 years. Control: Mean 23.1 ± 4.3 years. ExMD: Mean 22.6 ± 3.3 years	Sport: Distance runners. Athlete level: Competitive/trained. Country: USA	ExMD; amenorrhea	Pre-post-intervention comparison; eumenorrheic control group at baseline	24	Primary outcome: Resumption of menses. Secondary outcomes: Reproductive and thyroid hormones; EI; EA; BMD; bone content; muscle strength/power; protein metabolism; POMS	7 of 8 resumed menses, <8-month ExMD showed greater skeletal improvements; increased EA and EI (not statistically significant); minimal bone/muscle changes; mood improved slightly; ExMD preceded >8 mo had lower BMD
De Souza et al. [29] (2021)	Design: Randomised control trial Classification: Non-pharmacological (diet)	Exercising women (n = 76). Age range: 19–23 years. Oligo/Amen + Cal: Mean 21.3 ± 0.5 years. Oligo/Amen Control: Mean 20.7 ± 0.5 years	Multi-event (running, cycling, triathlon). Country: USA	REDs related to menstrual dysfunction (amenorrhea/oligomenorrhoea)	Standard diet vs. increased energy intake	48	Primary outcome: Recovery of menses. Secondary outcomes: LH pulse frequency; energy availability; estradiol	The increased energy intake group had a significantly higher rate of menstrual recovery (79%) and LH pulse frequency improvement than controls (29%)
De Souza et al. [30] (2022)	Design: Randomised control trial. Classification: Non-pharmacological (energy intake intervention)	Exercising women (n = 76) (oligo/amenorrhea + Cal: n = 40; Control: n = 36). Age range: 20–22 years. OV reference: Mean 23.4 ± 0.7 years. Oligo/Amen + Cal: Mean 21.3 ± 0.5 years. Oligo/Amen: Mean 20.7 ± 0.5 years	Endurance activities (running, cycling, swimming), ball sports, aesthetic sports, power sports. Country: USA	Oligo-amenorrhea; risk of REDs	Control group without intervention	52	Primary outcome: Bone mBMD. Secondary outcomes: Body mass/BMI/fat mass/fat %/IGF-1; menstrual frequency	Intervention increased weight, BMI, fat mass, fat %, IGF-1, and menstrual frequency (*p* < 0.05) vs. control; total body and spine aBMD: no change vs. control; femoral neck and hip aBMD decreased over 12 months in both groups
Dueck et al. [31] (1996)	Design: Intervention study with matched controls. Classification: Non-pharmacological	Female athletes (n = 4) (1 amenorrheic; 3 eumenorrheic comparators). Age range 19–20 years	Sport: Endurance running. Athlete level: Collegiate. Country: USA	Amenorrhea	Eumenorrheic controls	15	Primary outcome: Energy balance. Secondary outcomes: Body fat percentage, fasting LH and cortisol, menstrual function	Amenorrheic athlete regained normal hormonal levels and menstrual cyclicity with improved energy balance, increased fat mass, increased LH, and reduced cortisol. Control athletes showed stable hormones/fat
Fahrenholtz et al. [32] (2023)	Design: Non-randomised controlled trial. Classification: Non-pharmacological (lectures and counseling)	Female athletes (n = 50). Age range: 18–35 years. FUEL: Mean 24.1 ± 4.7 years. CONTROL: Mean 25.3 ± 4.8 years (32 in FUEL intervention; 18 in control group)	Sport: Endurance sports (long-distance running, orienteering, triathlon, cycling, cross-country skiing, biathlon). Athlete level: Recreational and competitive. Country: Norway, Sweden, Ireland, Germany	Risk of REDs (LEAF-Q ≥ 8)	Control group without lectures/counseling	16	Primary outcome: Sports nutrition knowledge (interviews and self-perceived). Secondary outcomes: Dietary behaviour (7-day weighed food record, habit questions)	Strong evidence of improved nutrition knowledge (28% increase); weak evidence of improved behaviour (e.g., carbohydrate intake days/week improved marginally)
Fredericson et al. [33] (2023)	Design: Intervention study. Classification: Non-pharmacological (nutrition education)	Collegiate distance runners (n = 56 retrospective phase; n = 78 intervention phase)	Sport: Distance running (≥800 m). Athlete level: Collegiate. Country: USA	At a high risk of bone stress injury (BSI)	Compare bone stress injury rates in the historical and intervention phases	208	Primary outcome: Incidence of BSI. Secondary outcomes: Energy intake, bone density	The intervention resulted in a significant reduction in BSI alongside improved energy intake and bone mineral density
Guebels et al. [34] (2014)	Design: Intervention with control group. Classification: Non-pharmacological (dietary supplement)	Active women (n = 17) (8 ExMD plus 9 eumenorrheic active controls). Age range 19–29 years	Sport: Endurance sports. Athlete level: Endurance trained (≥7 h/week). Country: USA	ExMD; eumenorrhea	eumenorrheic control at baseline	24	Primary outcome: RMR. Secondary outcome: E; TEE; EA; EB; weight	ExMD had higher baseline TEE and more negative EB; RMR was higher in ExMD; increased weight; no significant changes in EA, EB, or RMR post-intervention; EA ranged from 28.2–36.7 to 30–45.4 kcal/kgFFM/d
Kopp-Woodroffe et al. [35] (1999)	Design: Intervention pilot study. Classification: Non-pharmacological (diet and exercise)	Active females (n = 4). Age range: 19–34 years	Sport: Running, swimming, cycling, weight training. Athlete level: Competitive (training 7+ h/week). Country: USA	athletic amenorrhea	Pre-post-intervention comparison	20	Primary outcome: Energy balance. Secondary outcomes: EI, EE, serum biomarkers	Improved energy balance; increased body weight; increased body fat; improved nutrient status; resumption of menstrual function in 3 participants
Lagowska et al. [36] (2014)	Design: Non-randomised intervention. Classification: Non-pharmacological	Female athletes (n = 52) (31 athletes, 21 ballet dancers). Age range: 16–21 years. Ballet: Mean 17.1 ± 0.9 years. Athletes: 18.1 ± 2.6 years	Sport: Ballet and endurance sport (triathlon, rowing, swimming). Athlete level: Competitive/trained. Country: Poland	Menstrual disorders (amenorrhea/oligomenorrhea)	Baseline vs. post-intervention; dancers vs. athletes within-study	36	Primary outcome: Restoration of menses. Secondary outcomes: Energy and nutrient intake; TEE; EA; body composition; hormones (LH, FSH, E2, P, TSH, PRL, SHBG, leptin); resting metabolic rate (RMR)	Regular menses resumed in 10/52 participants; EA and energy intake increased; RMR was low in dancers; weight/BMI/body composition improved in dancers; not athletes; higher body fat linked to menstrual recovery
Łagowska et al. [37] (2014)	Design: Non-randomised intervention. Classification: Non-pharmacological	Female athletes (n = 31). Age range: 15–21 years. Mean: 18.1 ± 2.6 years	Sport: Rowing, synchronised swimming, and triathlon. Athlete level: Professional. Country: Poland	Amenorrhea; oligomenorrhea (low energy availability related)	Pre-post within-subject; no independent control group	12	Primary outcome: Energy intake. Secondary outcomes: TEE, EA, EB, body composition, LH, FSH, oestradiol, progesterone	Significant improvement in energy intake, EA, EB, and LH hormone profiles, but no restoration of menstrual cycles within 3 months
Mallinson et al. [38] (2013)	Design: Case report. Classification: Non-pharmacological (diet)	Exercising women (n = 2). Age range 19–24 years	Sport: Multi-event (running, weight training, rock climbing, hiking, downhill skiing, dancing). Athlete level: Recreational (exercise 7+ h/week). Country: USA	Functional Hypothalamic Amenorrhea (FHA)	Pre-post-case report	6.9	Primary outcome: Recovery of menses. Secondary outcomes: Energy intake; total body mass.	P1: +4.2 kg body mass (8%), P2: +2.8 kg (5%); both showed ↑ REE, ↑ T3 and leptin, ↓ ghrelin; menstrual resumption at day 23 (P1) and 74 (P2); ovulation and cycle regularity returned closely with weight gain
Meyer et al. [39] (2025)	Design: Clinical practice report. Classification: Non-pharmacological (consultation program)	Female Athletes (n = 58). Age range: 13–24 years. Mean: 18.3 ± 5.5 years	Sport: Multi-event (game sports, endurance, strength, martial arts, aesthetic, technical). Athlete level: Competitive (3–27 h/week). Country: Germany	Suspected REDs	Pre-post-intervention comparison	188	Primary outcome: Improvement in REDs symptoms. Secondary outcome: Energy intake; bone density	A multidisciplinary approach resulted in improvement in REDs symptoms and health outcomes
Michopoulos et al. [40] (2013)	Design: Randomised control trial. Classification: Non-pharmacological (cognitive behavioral therapy)	Women (n = 17). Age range: 23–27 years. Observation: Mean 25.1 ± 1.8 years. CBT: 25.3 ± 0.6 years	Sport: Not identified. Athlete level: Not identified. Country: USA	Functional Hypothalamic Amenorrhea (FHA)	Observation (no CBT)	20	Primary outcome: Cortisol. Secondary outcomes: Leptin, TSH, total/free T3 and T4 concentrations	CBT significantly lowered cortisol (*p* = 0.006) and increased leptin and TSH; the observation group showed no changes
Solstad et al. [41] (2025)	Design: Mixed methods; quasi-experimental. Classification: Non-pharmacological	Female athletes (n = 44). Age range 18–35 years. Evaluation questionnaire: Mean 24.6 ± 4.8 years. Qualitative interview: Mean 25.1 ± 5.8 years	Sport: Endurance (long-distance running, orienteering, cycling, triathlon, cross-country skiing, biathlon). Athlete level: Competitive (training 20+ sessions/month). Country: Multicentric (Germany, Ireland, Norway, Sweden)	Relative Energy Deficiency in Sport (REDs, at risk). Self-reported diagnosis/no clinical diagnosis	Digital lectures and consultation vs. digital lectures only	16	Primary outcome: Overall satisfaction. Secondary outcome: Energy, confidence, body satisfaction, knowledge	Participants reported high satisfaction and intention to recommend; no significant LEAF-Q group differences post-intervention, but improvements in menstrual and GI symptoms were seen at 6 and 12 months follow-up

Note: BMI—Body Mass Index; REDs—Relative Energy Deficiency in Sport; LH—luteinizing hormone; FSH—follicle-stimulating hormone; TSH—thyroid-stimulating hormone; PRL—prolactin; SHBG—Sex Hormone Binding Globulin; IGF-1—Insulin-like Growth Factor 1; T3—triiodothyronine; EA—energy availability; EB—energy balance; TEE—Total Energy Expenditure; RMR—resting metabolic rate; ExMD—exercise-associated menstrual dysfunction; ROM—restoration of menses; POMS—Profile of Mood States; ↑/↓ = increase/decrease in the subsequent variable/outcome.

**Table 2 sports-13-00453-t002:** Summary characteristics of pharmacological interventions.

Authors	Study Design and Classification	Sample and Sample Size	Sport, Athlete Level, and Geographical Location	Participant Identification	Comparator or Control	Intervention Duration (Weeks)	Outcome Measures	Results
Ackerman et al. [16] (2019)	Design: Randomised control trial. Classification: Pharmacological (Oestrogen)	Female athletes (n = 121) (40 patch, 41 pill, 40 none). Age range: 14–25 years. PATCH: Mean 19.86 ± 0.40 years. PILL: Mean 20.30 ± 0.44 years. NONE: Mean 19.35 ± 0.38 years	Sport: Weight-bearing endurance athletes. Level: Competitive. Country: USA	Oligo-amenorrhea	Transdermal estradiol + cyclic progesterone (PATCH); oral ethinyl estradiol-based pill (PILL); no oestrogen (NONE)	260	Primary outcome: Bone mineral density. Secondary outcomes: No further outcome measures identified.	The transdermal PATCH group had significantly greater increases in spine and femoral neck BMD Z-scores compared with PILL and NONE; hip BMD improved vs. PILL
Dadgostar et al. [42] (2018)	Design: Randomised clinical trial. Classification: Pharmacological (hormone therapy)	Female athletes (n = 18). (OCP n = 10, control n = 8). Age range 18–30 years	Sport: Multi-sports (34 different exercise fields). Athlete level: Elite. Country: Iran	FHMD	Control group (placebo or standard care without hormonal therapy)	36	Primary outcome: Bone mineral density. Secondary outcomes: Cardiovascular factors (cholesterol, LDL, HDL, triglycerides)	The intervention group (hormone therapy) demonstrated an increase in bone mineral density and improved cardiovascular profile (lower LDL, higher HDL) after 12 months of treatment, while the control group remained largely the same or even deteriorated in bone density
Gibson et al. [43] (1999)	Design: Randomised clinical trial. Classification: Pharmacological (HRT) and non-pharmacological (calcium)	Elite female athletes (n = 34) (HRT + Ca n = 10; Ca only n = 14; control n = 10). Age range 18–35 years	Sport: Middle and long-distance runners. Athlete level: Elite. Country: United Kingdom	Amenorrhea; ExMD	HRT + calcium vs. calcium alone vs. no treatment (control)	52	Primary outcome: BMD. Secondary outcomes: Changes in menses	EU group (resumed menses) saw significant BMD increases (~4% in spine and Ward’s triangle); AM group (persistent amenorrhea) experienced BMD losses. “Intention-to-treat” analysis: spine BMD increased only by 1.5% due to withdrawals and menses return in controls
Warren et al. [44] (2003)	Design: Clinical, placebo-controlled, randomized trial. Classification: Pharmacological (HRT + calcium)	Female dancers (n = 18). Age range: 17–27 years. Amenorrhea: Mean 20.8 ± 3.1 years (treated). Mean 19.2 ± 3.4 years (placebo). Mean 24.7 ± 4.6 years (control)	Sport: Ballet. Athlete level: Elite. Country: USA	Exercise-induced amenorrhea; delayed menarche	Hormone therapy (Premarin + Provera) + calcium vs. placebo + calcium	52	Primary outcome: Bone mineral density (foot, wrist, lumbar spine). Secondary outcomes: Resumption of menses	No BMD difference between HRT and placebo groups; five placebo participants who resumed menses showed BMD gains without normalization; Persistent osteopenia despite therapy, suggesting factors beyond estrogen depletion.

Note: BMD = Bone Mineral Density; OCP = oral contraceptive pill; HRT = Hormone Replacement Therapy; Ca = calcium; PATCH = transdermal 17β-estradiol patch with cyclic oral micronized progesterone; PILL = combined oral contraceptive pill (ethinyl estradiol + desogestrel); NONE = no hormonal therapy.

**Table 3 sports-13-00453-t003:** A summary table for non-pharmacological studies (using TIDieR sub-headings).

Authors	REDs Diagnosis Criteria Used	Control or Comparator	Intervention Strategy	What Was Provided (Materials/Information)	Intervention Setting (Where)	Duration and Dosage	Who Provided (MDT Involvement)
Arends et al. [27] (2012)	Measuring oestradiol (pg/mL), LH (mIU/mL), follicle-stimulating hormone (FSH; mIU/mL), thyroid-stimulating hormone (mIU/L), and prolactin (ng/mL); a urine pregnancy test	Within-sample comparison: “resumers” vs. “non-resumers” following non-pharmacologic intervention	Increased dietary intake and/or reduced exercise expenditure	Individualized nutrition and training recommendations to address low energy availability	University student health center	Mean time to restoration of menses: 15.6 ± 2.6 months; intervention varied per athlete	College health center providers
Cialdella-Kam et al. [28] (2014)	Exercise-associated menstrual dysfunction: ≥3 months of no menses or ≤6 cycles/year; identified as low EA from energy intake/expenditure logs	Eumenorrheic athlete controls assessed at baseline only	Daily carbohydrate-protein supplement (360 kcal/day: 54 g CHO + 20 g PRO) for 6 months	360 kcal CHO-PRO drink; baseline and ongoing 7-day diet and activity logs to track energy balance	University-based lab/clinical setting	6 months; supplement consumed daily; mid-intervention check at 3 months	Nutrition and exercise science research team
De Souza et al. [29] (2021)	Self-reported amenorrhea (>3 months) or oligomenorrhea (<7 cycles in 12 months or cycles of 36–89 days), with suppressed reproductive hormones; exercise ≥ 2 h/week	40 in + Cal intervention; 36 in control—maintenance of usual energy intake (control arm)	Increase energy intake by 20–40% above baseline needs + nutritionist guidance and supplements	Nutrition counseling; energy bars (220–300 kcal) and pre-measured nut servings; RMR/energy needs assessments; urinary hormone logs	University-based clinical research center	12 months	Registered clinical dietitians and clinical psychologists; the research team included sports endocrinology/investigators
De Souza et al. [30] (2022)	Self-reported cycles; secondary FHA due to energy deficiency. Confirmed via menstrual history and hormonal measures resembling prior REFUEL paper	Oligo/Amen control group maintaining usual diet/energy intake	Increased energy intake by 20–40% over baseline needs (around +352 kcal/d)	Personalized nutrition counseling plus supplemental energy sources to raise intake	Clinical nutrition research	12 months of continuous intervention	Research nutritionists and investigators
Dueck et al. [31] (1996)	estradiol, progesterone, luteinizing hormone (LH), follicle-stimulating hormone (FSH), and cortisol	3 matched eumenorrheic endurance athletes over the same 15-week period	Reduced training load + nutritional supplementation	360 kcal/day sport nutrition beverage; 1 additional rest day per week	University research facility	15 weeks; daily supplement; 1 added rest day/week	Exercise physiologists and nutrition scientists
Fahrenholtz et al. [32] (2023)	LEAF-Q ≥ 8 (low energy availability); EDEQ < 2.5 (low risk of eating disorders)	Control group (n = 18), same assessments without lectures or counseling	A combination of weekly online sports nutrition lectures (16 total) and biweekly individual athlete-centered nutrition counseling	Lecture materials (evidence-based content), telephone/online interviews, personalized nutrition counseling every 2 weeks	Digital delivery across Norway, Sweden, Ireland, and Germany	16 weeks total; 16 lectures and 8 counseling sessions across that period	Delivered by researchers and practicing sports nutritionists
Fredericson et al. [33] (2023)	Female athlete triad risk screening to identify those at risk for low EA and BSI	Historical control: retrospective data from 2010 to 2013; intervention group followed prospectively from 2016 to 2020	Nutrition education sessions targeting energy availability and bone-building nutrients; individualized follow-up for high-risk athletes	Team-based presentations on fueling and bone health; individual sessions for triad-risk athletes	Two NCAA Division I collegiate running programs—team and individual sessions on site	7 years overall: pilot 2013–2016; intervention implemented 2016–2020	Sports dietitians and sports medicine/research personnel led education; team coaches and athletic trainers involved
Guebels et al. [34] (2014)	Menstrual status was confirmed by the presence/absence of ovulation; amenorrheic (no menses > 90 days), oligomenorrheic (cycle intervals >35 days) or eumenorrheic (10–13 cycles/year or intervals of ~28 days)	Active eumenorrheic women (n = 9) served as controls	+360 kcal/day supplement for 6 months to restore energy balance	Daily 360 kcal supplement; 7-day diet and exercise logs to monitor intake and expenditure	University lab-based research setting	6 months; supplement consumed daily; RMR assessed at baseline and 6 months	Exercise physiologists/nutrition researchers
Kopp-Woodroffe et al. [35] (1999)	Self-report menstrual history; reproductive and thyroid hormone analysis (serum progesterone, serum estradiol)	Pre-post-intervention	Decrease EE (1 rest day/week) + increase EI (11 oz CHO/Pro/Fat + micronutrient supplement per day)	Dietary supplement; calibrated diet scale; measuring cups	Home environment and lab	20 weeks	Researchers
Lagowska et al. [36] (2014)	Secondary amenorrhea (≥6 months) or oligomenorrhea (>35-day cycles), low EA based on diet and exercise logs	Pre-post-intervention comparison	Individualized diet plan to increase energy and nutrient intake without altering training	Monthly dietitian consultations; tailored diet based on energy/nutrient needs; 7-day dietary and exercise records	University nutrition lab	9 months, with evaluations at 3 (same population not included in review analysis), 6, and 9 months	Dietitians and research team (nutrition and exercise specialists)
Łagowska et al. [37] (2014)	Amenorrhea or oligomenorrhea (NH subtyping); lab exclusion of prolactin, thyroid, ovarian, androgen, hCG abnormalities	Pre-post-intervention comparison	3-month individualized dietary intervention to increase energy and nutrient intake, without reducing training	Tailored diet plans increasing energy (~+234 kcal), protein (+8 g), carbs (+66.8 g), calcium, vit A/D/C/folate; 7-day dietary/exercise logs	University nutrition lab	3 months; moderate daily energy/nutrient increases	Dietitians and the exercise science research team
Mallinson et al. [38] (2013)	Amenorrhea ≥ 3 months; daily urinary excretion of E1G and PdG metabolites for a 28-day period	Pre-post-case report	Increased dietary intake 20–30% above baseline TEE	Personalized caloric increase via energy bars (250–300 kcal): +276 kcal/day for P1, +1881 kcal/day for P2; ongoing dietary monitoring and support	University laboratory/clinical research facility	Until menses resumed: 23 days for P1; 74 days for P2	Women’s Health and Exercise Lab research team; exercise physiologists and nutrition researchers
Meyer et al. [39] (2025)	REDs suspicion via CAT1 (BMI, body fat %, weight loss, menstrual disorders, bone health history)	Pre-post-comparison	Interdisciplinary outpatient consultation including sports medicine, gynecology, psychosomatic medicine, dietetics, and psychiatry	Medical records review, anthropometry, performance diagnostics, lab tests, dietary records, psychological and gynecological assessments; tailored treatment/referral pathway	University Hospital Tübingen outpatient clinics (sports medicine, gynecology, psychosomatic, psychiatry, dietetics)	Mean treatment duration ~15 months (up to 36); initial high intensity in 2–3 months, follow-ups 1–2 years; appointments ranged from 2–15 in sports med, up to 4 in psychiatry	sports medicine physicians, gynecologists, psychosomatic/child and adolescent psychiatrists, registered dietitians, and health educators
Michopoulos et al. [40] (2013)	FHA defined by >3 months of amenorrhea; exclusion of other causes via serum E2 and FSH; prolactin, testosterone, DHEA, TSH, and urinary hCG	Randomized to CBT intervention (n = ~8) or observation control (n = ~9)	Weekly cognitivebehavioral therapy sessions	Structured CBT program focused on stress, dieting/exercise beliefs, and lifestyle behaviors	Clinical research center at Emory University	Approximate 20 weekly sessions (~5 months) (number inferred from context)	Clinical psychologists specializing in CBT under research supervision
Solstad et al. [41] (2025)	LEAF-Q score ≥ 8 (risk of REDs); EDE-Q < 2.5 (low risk of eating disorder)	33 participants partook in digital lectures and consultation; 11 participants completed digital lectures only	Digital sports nutrition lectures and individual consultations	Weekly online lectures, bi-weekly individualised nutrition counseling via motivational interviewing, focused on REDs, nutrition intake, and menstrual health	Home environment—online lectures; Zoom video	16 weeks	Sports nutritionists

Note: EA = energy availability; REDs = Relative Energy Deficiency in Sport; CBT = cognitive behavioral therapy; LH = luteinizing hormone; FSH = follicle-stimulating hormone; TSH = thyroid-stimulating hormone; E2 = estradiol; P = progesterone; hCG = Human Chorionic Gonadotropin; DHEA = dehydroepiandrosterone; BMI = Body Mass Index; RMR = resting metabolic rate; TEE = Total Energy Expenditure; LEAF-Q = Low Energy Availability in Females Questionnaire.

**Table 4 sports-13-00453-t004:** Tider summary table for intervention studies.

Authors	REDs Diagnosis Criteria Used	Control or Comparator	Intervention Strategy	What Was Provided (Materials/Information)	Intervention Setting (Where)	Duration and Dosage	Who Provided (MDT Involvement)
Ackerman et al. [16] (2019)	Serum estradiol and FSH were used for confirmation; prolactin, testosterone, DHEA, TSH, and urinary hCG were measured to exclude other causes	Three arms: transdermal estradiol + cyclic progesterone (PATCH); combined oral contraceptive pill (PILL); no estrogen/progesterone therapy (NONE), all received Ca and vitamin D	Transdermal 17β-estradiol patch vs. oral combined OCP vs. none	PATCH: continuous transdermal 17β-estradiol + cyclic oral micronized progesterone PILL: ethinyl estradiol/desogestrel pill NONE: no hormone therapy All: calcium and vitamin D supplementation	Clinical/research setting at Massachusetts General Hospital and affiliated centers	12 months; PATCH continuous; PILL cyclic	Sports/endocrine physicians, dietitians, psychologists, and research staff
Dadgostar et al. [42] (2018)	Self-report; serum FSH, LH, thyroid-stimulating hormone, prolactin, free thyroxine, serum testosterone and dehydroepiandrosterone one, and an ultrasonographic study of the ovary	Low-dose combined OCP; control (no OCP); both groups received calcium + vitamin D	Low-dose combined oral contraceptive (30 µg ethinyl estradiol + 150 µg levonorgestrel) vs. no OCP control	Daily hormone tablet; both groups: 1000 mg calcium + 400 IU vitamin D daily; dietary guidelines based on exercise intensity/duration	Clinical/research center in Tehran, Iran	9 months; OCP daily; supplements daily; monthly telephone check-ins and tri-monthly clinic visits	Sports medicine physicians and endocrinologists
Gibson et al. [43] (1999)	Secondary amenorrhea confirmed; hormonal evaluation conducted to exclude other causes (e.g., estrogen, FSH, prolactin, testosterone, DHEA, TSH, hCG)	Three groups: (A) HRT+1000 mg calcium, (B) calcium only, (C) no treatment	Estrogen–progestin HRT; calcium supplementation vs. no treatment	Daily HRT + 1000 mg Ca; Ca-alone group took 1000 mg Ca	Outpatient/research clinic	12 months; HRT dosage per standard protocols; calcium 1000 mg/day	Medical team at British Olympic Medical Centre
Warren et al. [44] (2003)	Serum estradiol and FSH measured at visits; prolactin, testosterone, DHEA, TSH, and urinary hCG	Pre-post-intervention comparison	Hormone therapy (estrogen and progesterone)	Oral hormone therapy regimen prescribed; nutritional counseling	Clinical setting	Hormone therapy administered over 2 years	Endocrinologists and gynecologists

Note: PATCH = transdermal 17β-estradiol patch with cyclic oral micronized progesterone; PILL = combined oral contraceptive pill (ethinyl estradiol + desogestrel); NONE = no hormonal therapy; OCP = oral contraceptive pill; HRT = Hormone Replacement Therapy; Ca = calcium; MDT = multidisciplinary team; LH = luteinizing hormone; FSH = follicle-stimulating hormone; TSH = thyroid-stimulating hormone; hCG = Human Chorionic Gonadotropin; DHEA = dehydroepiandrosterone.

**Table 5 sports-13-00453-t005:** Identifying effect sizes (Cohen’s d) from Ackerman et al.’s [16] study on biomarkers.

Parameter	Time	PATCH vs. PILL	PATCH vs. NONE	PILL vs. NONE
Estradiol	6 m	1.692	3.495	−0.392
Estradiol	12 m	1.126	3.645	−0.252
SHBG	6 m	−12.053	1.892	12.772
SHBG	12 m	−9.33	−1.122	8.252

Note: The PATCH group was treated with physiological estrogen replacement via 100 mcg transdermal 17β-E2 and 200 mg cyclic micronized progesterone. The PILL group was treated with combined oral contraceptives of 30 µg ethinyl estradiol with 0.15 mg desogestrel.

**Table 6 sports-13-00453-t006:** Identifying effect sizes from Ackerman et al.’s [16] study on bone turnover markers.

Parameter	Time	PATCH vs. PILL	PATCH vs. NONE	PILL vs. NONE
IGF-1	6 m	2.103	−0.133	−0.778
IGF-1	12 m	3.168	−0.096	−3.063
P1NP	6 m	0.342	−0.033	−0.377
P1NP	12 m	0.291	0.085	−0.24

Note: The PATCH group was treated with physiological oestrogen replacement via 100 mcg transdermal 17β-E2 and 200 mg cyclic micronized progesterone. The PILL group was treated with combined oral contraceptives of 30 µg ethinyl estradiol with 0.15 mg desogestrel.

## Data Availability

Critical appraisal, narrative synthesis, and meta-analysis data are available upon request.

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

guideline for reporting systematic reviews. BMJ.

