# Peer review of "Pharmacological vs. Non-Pharmacological Treatment in the Management of Relative Energy Deficiency in Sport (REDs): A Systematic Review and Meta-Analysis"

_sports, 2025, doi:10.3390/sports13120453_

Round 1
Reviewer 1 Report
Comments and Suggestions for Authors
The manuscript addresses an important and timely topic, namely the comparative effectiveness of pharmacological versus non-pharmacological interventions for the management of relative energy deficiency in sport. The overall structure and logic of the paper are appropriate for a systematic review with meta-analysis, and the authors clearly have good command of the clinical area. At the same time, there are a number of issues in reporting, methodological transparency and language that, in my view, require substantial revision before the manuscript is suitable for publication.
Starting from the title, it is generally clear and informative. However, the choice of “Relative Energy Deficiency in Sport (RED-S)” in the title contrasts with the notation used throughout most of the text, where “REDs” appears frequently and sometimes inconsistently. I would recommend deciding on one convention, ideally aligned with the most recent consensus document, and then applying it consistently across title, abstract, text and figures. In addition, the title promises a direct comparison between pharmacological and non-pharmacological treatments; at present the review primarily presents the two streams in parallel rather than a true comparative synthesis. You might consider making this clear either in the title or in the abstract (e.g., “systematic review of pharmacological and non-pharmacological interventions”) unless you are able to strengthen the comparative element in the Results and Discussion.
The abstract is concise and generally well structured, but in its current form it remains somewhat generic and under-specific for a systematic review with meta-analysis. It would be useful to report more concrete details: for example, the exact total sample size, the main quantitative estimates from the meta-analyses (effect sizes and confidence intervals), and the key domains where evidence was judged as low versus moderate certainty. The description of data sources is appropriate, but you should correct minor spelling issues (e.g. “CINHAL” should be “CINAHL”) and ensure that the time frame is accurate and consistent with the Methods (search “until July 2025”). The conclusion section of the abstract is broadly in line with the results, but it could be sharpened by explicitly stating that the certainty of evidence is generally low to moderate and by clarifying in what “specific situations” pharmacological management appears appropriate. At present this is somewhat vague.
The introductory section provides a solid overview of the concept of RED-S and its evolution from the female athlete triad. The epidemiological data on prevalence and the description of multisystem consequences are appropriate and relevant. Nevertheless, there are several stylistic and grammatical issues that detract from readability. For instance, early in the introduction “Relative Energy Deficiency in Sport (REDs) a relatively new and evolving clinical model…” lacks the verb “is”, and similar omissions and typographical errors recur elsewhere (“bur risk masking” instead of “but risk masking”, “as used” instead of “was used”, “evidence effected by quality downgrades” where “affected” is meant). A thorough language edit by a fluent English speaker or professional language service is recommended.
In terms of content, the rationale for the review is clearly stated, particularly the gap regarding comparative evidence for pharmacological versus non-pharmacological management. It may be helpful to temper the claim that “no systematic review or meta-analysis has directly compared” these approaches by qualifying it (“to our knowledge…”) and, ideally, by briefly summarising the scope of any previous related reviews to better position the present work. You already cite important previous narrative and descriptive work; a short paragraph distinguishing the present review more explicitly from these would make the contribution clearer.
The Methods section follows PRISMA conventions reasonably well, and registration on PROSPERO is a strength. However, some elements of reporting need clarification or expansion to meet current standards for systematic reviews and meta-analyses.
Regarding eligibility criteria, the use of the PICOS framework is appropriate, and the inclusion/exclusion criteria for participants are generally well described. Restricting the sample to females and excluding male athletes is defensible, given the current evidence base, but this should be better justified in terms of the review’s aims, especially since RED-S is now conceptualised as affecting both sexes. At minimum, this choice and its implications for generalisability should be explicitly acknowledged in the Limitations and in the EDI section. There are also minor wording issues that should be corrected, for example “Studies were also excluded is participants had a clinical diagnosis” should read “if participants had…”.
The description of the interventions is broadly clear, but the grouping into “pharmacological” and “non-pharmacological” arms could be better standardised. For instance, some studies combined hormonal therapy with calcium and vitamin D supplementation, which you classify as pharmacological, whereas calcium alone appears sometimes under the non-pharmacological umbrella and sometimes within pharmacological trials. It would be useful to specify a priori operational definitions for each category and apply them consistently, perhaps in a short paragraph at the start of the Intervention subsection. Similarly, the exclusion of studies focusing exclusively on psychological measures is understandable, but it may be worth reconsidering this, given the recognised psychological dimensions of RED-S; at least a justification should be added.
The Outcomes section describes an appropriately broad range of endpoints (menstrual resumption, EA, body composition, hormonal markers, physical function, psychological measures). However, at present the primary versus secondary outcome hierarchy is not entirely clear at the review level. You report primary outcomes at the individual study level, but for the meta-analyses it would be valuable to state explicitly which outcomes were designated as primary for the review and how this related to the choice of domains for quantitative synthesis. If only certain physiological domains (e.g. fat mass, body fat percentage, TT3) were judged suitable for pooling, the criteria for that decision should be specified.
The description of the search strategy is fairly detailed with regard to databases, search terms and grey literature sources, which is positive. There are, however, a few improvements that would strengthen transparency. The exact search strings (including Boolean operators, truncations and limits) are not reported in full; the journal will typically expect these either in the main text or as a supplementary file. It would also be important to state whether any language restrictions were applied in practice, despite the statement that no restriction was made, and how non-English papers were handled. The description “first 30 pages” for some search engines is somewhat unusual; this could be clarified (e.g. whether “pages” refers to pages of results).
Study selection and data extraction procedures are clearly described, including independent screening by two reviewers and use of Covidence. It would be helpful to report inter-rater agreement (e.g. kappa statistics) or at least the proportion of disagreements resolved by consensus, if available. The description of the extraction tool is fairly detailed, but the reference to “Tidier guidelines” appears to be a typographical error; it should be “TIDieR”. It may also be worth indicating whether the extraction form was piloted and refined before full use.
The risk of bias assessment is one of the strengths of the work, as the appropriate tools (ROB-2, ROBINS-I, JBI) are used according to design. However, the reader currently has limited visibility of the results, which are relegated to a supplementary file. Given that low study quality is a key factor influencing your certainty ratings, I would suggest summarising the risk-of-bias patterns in the main text, perhaps in a brief paragraph and/or a condensed table (e.g. number of studies at low, some concerns, high risk by domain). This would also facilitate understanding of how these judgements informed the GRADE assessments.
The statistical analysis section is too concise at present and needs to be expanded. The statement that “mean differences were considered where possible” does not provide enough information. It is important to specify:
– Whether fixed-effect or random-effects models were used for the meta-analyses and the rationale for this choice.
– How heterogeneity was quantified (e.g. I², τ²) and whether any thresholds were used to interpret it.
– How standard deviations of change scores were handled when not reported (you mention in the Discussion that an SD was estimated for one study; the formula and assumptions should be stated in Methods).
– How you dealt with multi-arm studies and multiple outcomes from the same study to avoid double counting.
– The statistical software used for analyses.
In addition, the labelling of figures is currently confusing. The figure referred to as “Figure 1” in the Results is captioned as a meta-analysis of fat mass, whereas in PRISMA-compliant papers Figure 1 would usually be the flow diagram. This needs to be corrected so that the numbering and captions match the content. Similarly, the meta-analytic forest plots need to be clearly presented with study names, effect sizes, confidence intervals and weight, which is difficult to assess from the current text alone. Please ensure that figures are of sufficient resolution and that all abbreviations in the figures are explained in their legends.
The narrative synthesis of results is generally clear and organised by outcome domain, which works well. The sections on menstrual function recovery, EA, body composition and biomarkers provide useful quantitative summaries (proportions of studies and participants improving, ranges of effect sizes and p-values) and the GRADE summaries are appreciated. However, there are a few areas where the narrative could be improved.
First, the meta-analytic findings should be described more explicitly. For the body fat percentage and fat mass analyses, it would be important to report the pooled mean difference (with 95% CI), the number of studies and participants included, and the heterogeneity statistics, rather than only stating that the evidence is “moderate” or “supported by meta-analysis”. The same applies to the TT3 analysis, where you note that the meta-analysis actually favoured the control group: this is a critical result that deserves clearer emphasis and interpretation in the Results, not only in the certainty assessment.
Second, while the separation of non-pharmacological and pharmacological evidence is sensible, the comparative dimension promised in the aims is only briefly touched on. It would be useful to include at least one small paragraph in the Results explicitly contrasting, at a high level, the patterns of effect and certainty between the two approaches, even if a formal indirect comparison was not feasible.
Third, for pharmacological interventions, the proportions of studies showing significant benefits on BMD, hormonal profiles and menstrual recovery are informative, but here again, providing more concrete figures from key trials (e.g. magnitude of BMD increase in the transdermal oestrogen group versus controls) would strengthen the section.
The Discussion shows good depth in linking findings to physiological mechanisms, particularly in the sections on hypothalamic–pituitary–ovarian axis suppression, energy availability thresholds, and the masking effects of exogenous hormones on menstrual recovery and bone health. These mechanistic explanations help to contextualise why non-pharmacological strategies targeting EA are foundational. That said, the Discussion would benefit from some restructuring and tightening.
At present, some paragraphs repeat content from the Introduction (for instance, general statements about the prevalence and multifactorial nature of RED-S) that could be shortened or removed. In contrast, some important methodological and interpretative issues are only briefly mentioned. I would suggest the following enhancements:
– Expand the paragraph that discusses heterogeneity and methodological limitations to be more specific. For example, elaborate on the variability in diagnostic criteria for RED-S/FHA, inconsistency in definitions of menstrual recovery (bleeding versus ovulation), small sample sizes, and the predominance of university-based cohorts from high-income countries. These factors have direct implications for generalisability and should be clearly articulated.
– When discussing the masking effect of hormonal therapies, you already provide an insightful critique. It would be valuable to link this more explicitly to your own review data, for example by highlighting the discrepancy between improvements in BMD under oestrogen therapy and the lack of endogenous menstrual recovery in certain trials, and how this influenced your GRADE ratings.
– The implications for practice section is useful, but in its current form it is partly presented as bullet-type statements in the manuscript draft. Recasting this into continuous prose would improve coherence and fit with the journal’s style. You can still convey the three or four main clinical messages in paragraph form while avoiding list formatting.
– It might also be helpful to add a short paragraph on implications for research design, beyond the call for “further research”. For instance, you could suggest priorities such as larger multi-centre trials, inclusion of male athletes, standardised RED-S diagnostic criteria and outcome sets, and longer follow-up for bone and reproductive endpoints.
The Conclusion section is concise but somewhat understated and could do more to directly answer the review question. It would be appropriate here to clearly restate that non-pharmacological interventions, predominantly dietary energy restoration and training adjustment, show consistent benefits on clinically relevant outcomes but that the certainty of evidence is often low to moderate due to methodological issues. Conversely, pharmacological interventions appear to improve BMD and some hormonal parameters but may not address the underlying low EA and can obscure true recovery. Making this contrast more explicit and reiterating your recommendation that pharmacological options should generally be reserved for specific indications and as adjuncts, rather than first-line therapies, would help readers.
From a formal perspective, there are several technical and stylistic points to address throughout the manuscript. The use of abbreviations is sometimes inconsistent; for example, EA, LEA, BMD, TT3, LH, FHA and RED-S/REDs are all used, but not always defined at first mention or used consistently thereafter. A careful pass to harmonise abbreviations and ensure each is defined only once at first use would be helpful. There are also a number of minor typographical errors, redundant spaces, and punctuation inconsistencies (e.g. stray periods, inconsistent use of hyphens and en dashes in ranges). These should be corrected systematically.
Finally, regarding the reference list, the authors have clearly made an effort to include up-to-date and relevant literature, including the most recent consensus statements and key clinical guidelines, which is commendable. Nevertheless, the formatting of references appears inconsistent. Some entries contain duplicated citation text within the same reference, some URLs are presented separately without full citation details, and there are slight inconsistencies in journal name presentation, volume and page formatting, and DOI notation. In addition, one reference appears to list two identical citations in succession. I recommend carefully checking the entire reference list against the journal’s style guide, removing duplicates, harmonising formatting, and ensuring that all in-text citations match the bibliography numeration. It would also be worth verifying that the claim about the absence of prior systematic reviews is consistent with the references cited.
In summary, this manuscript addresses a clinically relevant question and has a number of strengths, including a registered protocol, an explicit use of GRADE and an attempt at quantitative synthesis. However, important aspects of the methods reporting, figure labelling, statistical description, language quality and reference formatting need to be improved, and the comparative dimension between pharmacological and non-pharmacological interventions should be made clearer. For these reasons, my overall recommendation is that the manuscript requires major revisions before it can be considered for publication.
Reviewer 2 Report
Comments and Suggestions for Authors
Congratulations on the manuscript. I would like to address some inconsistencies and questions that arise from the article.
-
The registration number CRD420251073240 does not exist when searched in the PROSPERO database. In addition, both the abstract and the methods section state that the search was conducted until “July 2025” and that the PROSPERO record was registered on 17 June 2025.
-
This is a very limited meta-analysis, as it includes very few articles, and the contribution made with these data is quite weak.
-
The PRISMA flowchart figure is missing. The text indicates that it is presented in Figure 1, but Figure 1 is actually a forest plot.
-
In the inclusion and exclusion criteria, the manuscript states: “Case studies, observational studies, descriptive research were excluded...” However, these types of studies are later included in the tables.
-
For what the article contributes, the discussion is insufficient and the conclusion is overly strong.
-
More recent references are needed.
-
The tables are highly heterogeneous across studies and excessively long (making them difficult to read).
-
The article states that 19 studies were included, but in the text different and contradictory numbers appear.
-
Could you clarify why you include studies on FHA, LEA or exercise-related menstrual dysfunction as if they were equivalent to RED-S, when they are not the same diagnosis?
Round 2
Reviewer 1 Report
Comments and Suggestions for Authors
The authors have adequately and comprehensively addressed all major points raised during peer review. The methodology is now clearly described, including search strategy, study selection, risk of bias assessment, and the use of GRADE and TIDieR, in line with current reporting standards. The comparative focus between pharmacological and non-pharmacological interventions has been made explicit throughout the manuscript, and the conclusions are balanced, clinically meaningful, and consistent with the presented evidence. The rationale for including only female athletes and the limits to generalizability are appropriately discussed, together with clear implications for practice and future research. Remaining issues are purely minor (typographical and stylistic) and can be handled at the copy-editing stage. In my view, the manuscript is suitable for acceptance in its current revised form.
Reviewer 2 Report
Comments and Suggestions for Authors
Thank you for addressing the issues raised.